# Unfolding Protein-Based Hapten Coupling via Thiol–Maleimide Click Chemistry: Enhanced Immunogenicity in Anti-Nicotine Vaccines Based on a Novel Conjugation Method and MPL/QS-21 Adjuvants

**DOI:** 10.3390/polym16070931

**Published:** 2024-03-28

**Authors:** Ying Xu, Huiting Li, Xiongyan Meng, Jing Yang, Yannan Xue, Changcai Teng, Wenxin Lv, Zhen Wang, Xiaodan Li, Tiantian Sun, Shuai Meng, Chengli Zong

**Affiliations:** 1School of Marine Biology and Fisheries, Hainan University, Haikou 570228, China; 21210707030018@hainanu.edu.cn; 2Key Laboratory of Tropical Biological Resources of Ministry of Education, School of Pharmaceutical Sciences, Hainan University, Haikou 570228, Chinazhenwang202402@163.com (Z.W.);; 3School of Biotechnology, Jiangnan University, Wuxi 214126, China

**Keywords:** immunopharmacotherapy, anti-nicotine vaccine, unfolding protein, organic phase conjugation, thiol–maleimide click chemistry

## Abstract

Vaccines typically work by eliciting an immune response against larger antigens like polysaccharides or proteins. Small molecules like nicotine, on their own, usually cannot elicit a strong immune response. To overcome this, anti-nicotine vaccines often conjugate nicotine molecules to a carrier protein by carbodiimide crosslinking chemistry to make them polymeric and more immunogenic. The reaction is sensitive to conditions such as pH, temperature, and the concentration of reactants. Scaling up the reaction from laboratory to industrial scales while maintaining consistency and yield can be challenging. Despite various approaches, no licensed anti-nicotine vaccine has been approved so far due to the susboptimal antibody titers. Here, we report a novel approach to conjugate maleimide-modified nicotine hapten with a disulfide bond-reduced carrier protein in an organic solvent. It has two advantages compared with other approaches: (1) The protein was unfolded to make the peptide conformation more flexible and expose more conjugation sites; (2) thiol–maleimide “click” chemistry was utilized to conjugate the disulfide bond-reduced protein and maleimide-modified nicotine due to its availability, fast kinetics, and bio-orthogonality. Various nicotine conjugate vaccines were prepared via this strategy, and their immunology effects were investigated by using MPL and QS-21 as adjuvants. The in vivo study in mice showed that the nicotine–BSA conjugate vaccines induced high anti-nicotine IgG antibody titers, compared with vaccines prepared by using traditional condensation methods, indicating the success of the current strategy for further anti-nicotine or other small-molecule vaccine studies. The enhancement was more significant by using MPL and QS-21 than that of traditional aluminum adjuvants.

## 1. Introduction

Tobacco use, which has a great correlation to cardiovascular disease, lung disease, and tobacco-related cancer, is responsible for nearly eight million deaths and immeasurable economic loss annually all over the world. Smoking cessation remains challenging in spite of the use of nicotine replacement therapies, receptor antagonists, vaccines, or behavioral interventions [1,2]. Vaccines against nicotine were supposed to induce nicotine-specific antibodies that could decrease the concentration of nicotine in the blood circulation by binding nicotine molecules so that the distribution of nicotine in the brain would be reduced [2]. To cure or prevent nicotine addiction, a robust titer and consistent production of high-affinity anti-nicotine IgG was necessary [2]. However, as a small molecule, the immunogenicity of nicotine is poor. The promotion of the immune response and the enhancement of antibody titers and affinity are key factors for anti-nicotine vaccine development. The most utilized strategy is to covalently attach nicotine to protein carriers (KLH, CRM197, DT, TT, etc.) [3,4,5,6,7,8], which would be digested to shorter peptides and then presented by antigen-presenting cells (APCs) to activate helper T cells. Another effective approach for improving vaccine efficacy is adopting immune adjuvants that could stimulate the immune system. Besides traditional aluminum salts, many immunostimulants, such as CpG, MPL, Poly(I:C), etc., [3,4,5,6,7,8,9,10] were investigated for their enhancement function on immunization outcomes in anti-nicotine vaccine studies. It was proven that these adjuvants could undeniably improve the promotion of the secretion of antibodies and immunological cell factors, but the functions varied greatly by case. Despite numerous efforts devoted to anti-nicotine vaccine development and a great number of promising results in preclinical studies, there is still no licensed vaccine against nicotine [1]. Several clinical phase studies have already failed [11]. Most of the past efforts have been well reviewed. Some recent developments involve the use of nanoparticle and liposome systems to formulate vaccines [12,13,14,15,16,17], using a covalent nicotine–adjuvant attachment strategy [9], or the use of epitope fragments instead of whole protein for vaccine design [18]. Different administration approaches were also attempted [3].

Future studies on nicotine vaccine development remain challenging. However, generally, the development of new kinds of vaccines still focuses on two aspects: the structure of antigens and the use of adjuvants. We have been interested in the development of hapten–protein conjugation methodology for years. Recently, we developed the fast conjugation of a polysaccharide antigen and carrier protein in an organic solvent (dimethylformamide), and the product demonstrated great immunogenicity [19]. We wondered whether such a strategy can be used for a small-molecule, carrier protein conjugate. Furthermore, can thiol groups on a protein be used as conjugation sites (Figure 1)?

In this study, the conjugation of small molecules (modified nicotine antigen) with unfolding proteins through the Michael addition of disulfide reduction resulting in thiol–maleimide was successful. The reaction was conducted in an organic solvent, where small molecules like nicotine have good solubility, so the reaction rate was fast. We believe that this method may provide a more effective approach toward the preparation of anti-nicotine vaccines with a higher titer of antibodies as the conformation of protein becomes more flexible, and therefore more epitopes may become accessible and conjugated. We also investigated the adjuvant effect of the combination of MPL and QS-21, which are the main components of AS01 licensed by GSK in recent years [20], in order to explore a more effective vaccine formulation. Herein, we describe the synthesis of several conjugate anti-nicotine vaccines as well as the results of immunological studies.

## 2. Materials and Methods

### 2.1. Materials, Reagents, and Animals

All chemicals were purchased from Sigma-Aldrich (St. Louis, MO, USA), Acros (Chicago, IL, USA), and Thermo Fisher (Waltham, MA, USA). The chemicals were used directly without further purification unless noted. CRM197 carrier protein was purchased from FinaBio (Rockville, MD, USA). All chemicals and solvents were of analytical grade and used as received from commercial resources. Maxisorp 96-well ELISA plates were purchased from Sangon Biotech (Shanghai, China). BALB/c mice 6–8 weeks of age were used for immunological studies, which were purchased from the Sibeifu company.

### 2.2. Chemical Synthesis

Nicotine-derived hapten **5** and hapten **6**, ready for conjugation with protein, were synthesized following the routes shown in Figure 1.

Compound **5** was synthesized from nicotine **1** according to the literature [7].

Compound **6**: Under the condition of DIPEA and HATU, the condensation between **5** and commercially available 3-maleimidopropionic acid gave hapten **6** in DMF. The detailed procedures and characterization are described in Supporting Information.

### 2.3. Protein Formulation

For conjugation performed in DMF, carrier proteins and additives were appropriately formulated to assist the dissolution.

For BSA formulation, a 20 μL solution comprising 50 mg/mL of BSA, 100 μL of 10% imidazole (pH 7.2 adjusted by HCl), and 880 μL of water was prepared and lyophilized.

For CRM197 formulation, a 91 μL solution comprising 11 mg/mL of CRM197, 100 μL of 10% imidazole (pH 7.2 adjusted by HCl), and 810 μL of water was prepared and lyophilized.

### 2.4. Conjugation

#### 2.4.1. Conjugation in Buffer

Succinic acid-modified CRM197 (Suc-CRM197) was prepared according to the following procedure: A solution of CRM197 (~2 mg/mL) in 0.1 M Tris buffer (pH 8.65) was treated with succinic anhydride (1.0 M in DMSO, 3000 equiv.) at room temperature for 1 h. The filtrate was transferred to a pre-rinsed 15 mL 30 kD MWCO Amicon filter, filled with 0.1 M MES buffer (pH 5.80), and centrifuged at 4500× *g* at 20 ℃. A solution of the resulting Suc-CRM197 (~5 mg/mL) in a pH 5.80 MES buffer was treated with nicotine-derived hapten **1** (0.2 M in H_2_O, 800 equiv.) and EDC (2.0 M in H_2_O, 1000 equiv.) at room temperature for 20–24 h. The filtrate was transferred to a pre-rinsed 15 mL 30 kD MWCO Amicon filter, filled with 10 mM PBS buffer (pH 7.4), and centrifuged at 4500× *g* at 20 °C to remove excess reagents. The incorporation number of nicotine per CRM197 molecule was 18, which was determined by the MALDI-TOF-MS. This conjugate is referred to as CRM197-Nicotine(18) in the following text.

Succinic acid-modified BSA (Suc-BSA) was prepared according to the following procedure: A solution of BSA (~5 mg/mL) in 0.1 M Tris buffer (pH 8.65) was treated with succinic anhydride (1.0 M in DMSO, 3000 equiv.) at room temperature for 1 h. The filtrate was transferred to a pre-rinsed 15 mL 30 kD MWCO Amicon filter, filled with 0.1 M MES buffer (pH 5.80), and centrifuged at 4500× *g* at 20 °C until the final volume reached about 100 μL. A solution of Suc-BSA (~5 mg/mL) in a pH 5.80 MES buffer was treated with hapten **1** (0.2 M in H_2_O, 800 equiv.) and EDC (2.0 M in H_2_O, 1000 equiv.) at room temperature for 20–24 h. The filtrate was transferred to a pre-rinsed 15 mL 30 kD MWCO Amicon filter, filled with 10 mM PBS buffer (pH 7.4), and centrifuged at 4500× *g* at 20 °C to remove excess reagents. The incorporation number of nicotine per BSA molecule was 77, which was determined by the MALDI-TOF-MS. This conjugate is referred to as BSA-Nicotine(77).

#### 2.4.2. Conjugation in DMF

The lyophilized CRM197 was dissolved in 1 mL DMF and then treated by vortex and sonicate until no large particle was observed. After adding 100 μL of water, the mixture was treated by vortex and sonicate, followed by the addition of 1 μL of PS20 and 2 μL of trimethylphosphine. After 0.5 h, 1 mg of nicotine antigen **2** was added. After 5 h, the solution was diluted by adding 19 mL of a 50 mM PBS buffer solution (pH 6.5) to reduce the DMF percentage to less than 5%. Then, the reaction mixture was transferred to a pre-rinsed 15 mL 30 kD MWCO Amicon filter, filled with the 50 mM PBS buffer solution (pH 6.5), and centrifuged at 4500× *g* at 20 °C until the final volume reached about 250 μL. The filtration step was repeated twice with a 10 mM PBS buffer solution (pH 6.5). The residue was pipetted to a 2 mL vial. Two portions of 200 μL of 10 mM PBS buffer solution (pH 6.5) were used to rinse the filter, and the mixture was transferred to the vial. MALDI-TOF-MS showed that the incorporation number per CRM197 molecule was 4. This conjugate is referred to as CRM197-Nicotine(4).

According to the above protocol, by using 4.8 mg and 6 mg of antigen **2**, BSA-based nicotine conjugates BSA-Nicotine(24) and BSA-Nicotine(35) were prepared, respectively.

### 2.5. Adjuvants

MPL and QS-21 liposomes were provided by Shanghai Ayk Biotechnology Co., Ltd. (Shanghai, China). The concentrations were 0.215 and 2.222 mg/mL, respectively.

### 2.6. Immunological Test

#### 2.6.1. Vaccine Formulation

Nicotine–protein conjugates, i.e., CRM197-Nicotine(18), CRM197-Nicotine(4), BSA-Nicotine(24), BSA-Nicotine(35), and BSA-Nicotine(77), were kept in stock solution before use. The original concentrations were 1 mg/mL, 5 mg/mL, 2 mg/mL, 1 mg/mL, and 3.4 mg/mL, respectively. These candidates were mixed with adjuvants, i.e., MPL/QS-21 or Al(OH)_3_, and then diluted by physiological saline to certain volumes to prepare the formulated vaccines, where the nicotine contents were around 20 μg/mL. The concentrations of MPL, QS-21, and Al(OH)_3_ in all the formulations were 100 μg/mL, 50 μg/mL, and 1000 μg/mL, respectively.

#### 2.6.2. Immunization

All mice were used according to the animal ethics guidelines. The mice were immunized by intravenous administration with 100 μL of the formulated vaccines on days 1, 15, and 29. The mice were bled before each injection on days 0, 14, 28, and 42. One group of 10 mice (5 females and 5 males) was used as a negative control and was only treated with 0.1 mL of physiological saline on days 1, 15, and 29. Mouse blood samples were collected by orbital sinus bleeding for antibody titer determination.

#### 2.6.3. IgG Titer Determination

A 96-well microtiter plate was first coated with BSA-Nicotine(77), which had been dissolved in a 10 mM PBS buffer (pH 7.4). Each microwell contained 0.1 µg of Nic-BSA in 100 µL of 0.1 M PBS buffer (pH 7.4) solution. Then, the plate was kept at 4 °C overnight and washed 3 times with PBST (0.05% Tween-20 in PBS). The solution of 2% (*w*/*v*) BSA in PBS (pH 7.4) was added to each well, and the plate was incubated at room temperature for 2 h. The plate was treated with serially diluted sera (100 μL/well), incubated for 2 h at room temperature, and then washed (3 times) with PBST. A diluted solution (1:5000) of HRP-conjugated goat anti-mouse IgG, IgM, IgG1, IgG2a, IgG2b, or IgG3 in 1% BSA/PBST/pH 7.4 (100 µL per well) was added to each microwell, respectively. After incubation at room temperature for 1 h, the plate was finally washed (5 times by PBST), followed by the addition of a TMB solution to the wells (100 µL per well). The plate was placed in the dark for 10 min for color development and terminated by a solution of 1 M HCl (100 µL per well). The OD value was then recorded at 450 nm. The antibody titer was determined as the highest dilution indicating a 0.1 absorbance, after subtracting the background. Pre-immunization sera (day 0) were used as the negative control.

### 2.7. Clinical Observations

Mice were observed daily and monitored for signs of distress or vaccine-associated toxicity, which included changes in gait, posture, and behavior. As part of basic toxicity-associated measurements, the mice were weighed weekly starting from day 0 for 8 weeks.

## 3. Results

### 3.1. Preparation of Anti-Nicotine Vaccines

To prepare the nicotine and protein conjugate vaccines, we first synthesized maleimide (Compound **6**) and amine-functionalized nicotine (Compound **5**). Commercially available nicotine was firstly functionalized by bis(pinacolato)diboron catalyzed by [Ir(COD)OMe]_2_ [21]. Interestingly, the air-stable precatalyst Ir(COD)(Phen)Cl was first used for C-H borylation [22], the yield of which was found unideal, so the catalyst was switched to [Ir(COD)OMe]_2_. Then, the hydroxyl group of Compound **3** was reacted with *tert*-butyl *N*-(2-bromoethyl)carbamate followed by *tert*-butyl group removal to reveal the amine group, yielding Compound **5**. It was reacted with 3-maleimidopropionic acid to yield the target Compound **6**.

Five conjugated anti-nicotine vaccine candidates were prepared, three of which used BSA as a carrier, and the other two used CRM197.

CRM197-Nicotine(18) and BSA-Nicotine(77) were assembled by amino-carboxylic acid condensation in an aqueous phase (Figure 2). First, CRM197 or BSA was treated with succinic anhydride in an alkaline Tris buffer to acylate the amino groups on the Lys, Arg, or N-terminal of the proteins, resulting in more carboxylic acids on the protein ready for condensation. Then, hapten **5** was conjugated to Suc-CRM197 or Suc-BSA using EDC in an MES buffer to obtain CRM197-Nicotine(18) and BSA-Nicotine(77).

In our strategy (Figure 3), the proteins were initially dissolved in water with 5% sucrose. After lyophilization, the cake was then dissolved in DMF following our previously reported procedure (2022 Carbohydrate Polymers). Then, the solution was treated using trimethylphosphine, a strong reductant that can reduce disulfide to free thiol rapidly at room temperature. The reduction was accomplished in DMF with a small amount of water (10%, *v*/*v*) and (1 μL) PS20 to assist in the dissolution of the protein. The resulting thiols were subjected to the click reaction with maleimide-derived hapten **6**, which did not call for extra solvent to assist the dissolution as DMF accounted for the majority in the reaction mixture. Thus, CRM197-Nicotine(4), BSA-Nicotine(24), and BSA-Nicotine(35) were prepared with our strategy (Figure 3).

The average loading number of nicotine in each conjugate was determined by MALDI-TOF-MS (Table 1, Figure 2). As can be seen from the data, the succinic anhydride derivatization of both proteins could ensure a good number of nicotines coupled on the protein (18 and 77 for CRM197 and BSA, respectively). For the unfolding protein approach, by controlling the amount of nicotine added to the reaction, two nicotine-conjugated vaccines were synthesized (BSA-Nicotine(24) and BSA-Nicotine(35)). According to multiple studies [5], a low amount of nicotine on a carrier protein is not ideal for vaccine immunogenicity, resulting in the vaccine being not synthesized. CRM197 has two disulfide bonds, so only four Compound 6 could be conjugated to generate CRM-Nicotine(4).

With these five conjugates in hand, six vaccine candidates were formulated by mixing the conjugates with adjuvants. All five conjugates were formulated with MPL/QS-21 to immunize the mice. Besides these five candidates, the BSA-Nicotine(77) conjugate was also formulated with aluminum hydroxide in order to compare the adjuvant effect of MPL/QS-21 to that of Al(OH)_3_ (Table 2). Prior to formulation, protein concentrations were determined using the BCA method. Each dose contained 5 μg of MPL, 10 μg of QS-21, and 2 μg of the nicotine antigen.

### 3.2. Immunological Effect

The mice were randomly divided into groups of ten and immunized with the above six vaccines via subcutaneous injection according to the schedule shown in Figure 3. The negative control group was treated with physiological saline instead of vaccines but bled together with the test groups. The mice treated with physiological saline, CRM197-Nicotine(18), CRM197-Nicotine(4), BSA-Nicotine(24), and BSA-Nicotine(35), were weighed before immunization and once every three days during the vaccination protocol to monitor their growth for the associated weight loss possibly related to vaccine-resulted toxicity. The weight-changing curves are demonstrated in Figure 4, which were largely unchanged over the course of experiments.

The titers of the anti-nicotine IgG antibodies in the sera obtained on days 14, 28, and 42 from the mice immunized with the six vaccines were determined by ELISA assays, and the results are shown in Figure 5A–C, respectively. No significant IgG was observed in the mice treated with CRM197-Nicotine(4). The titers of IgG antibodies induced by CRM197-Nicotine(18) and BSA-Nicotine(77) were at much higher levels than the other vaccines after the first immunization. However, after the first boost vaccination, the IgG antibodies in the mice immunized with BSA-Nicotine(24) and BSA-Nicotine(35) increased rapidly (~100-fold increase compared with the first injection). Finally, in the anti-sera obtained on day 42, the titers of IgG in the mice immunized with CRM197-Nicotine(18) and BSA-Nicotine(24) reached a high level of about 1 × 10^6^. The IgG antibodies stimulated with BSA-Nicotine(35) became more than those of BSA-Nicotine(77) as well. The IgG titers in the mice treated with aluminum adjuvant were also significant but obviously at a much lower level than the other vaccines except for CRM197-Nicotine(4).

The patterns of IgG subclasses in the mice vaccinated with CRM197-Nicotine(18), BSA-Nicotine(24), BSA-Nicotine(35), and BSA-Nicotine(77)/Al were also investigated (Figure 6). Generally, the titers of all IgG subclasses induced by BSA-Nicotine(77)/Al were at a significantly lower level than the other candidates, indicating the high efficacy of the MPL and QS-21 adjuvant system. The adjuvant system seems to assist the generation of a balanced level of IgG subtypes, as can be seen in Figure 6A–D. For the unfolded BSA-based conjugate, BSA-Nicotine(18) and BSA-Nicotine(24) generated similar IgG1 levels to other vaccines. BSA-Nicotine(18) seems to have generated more balanced IgG subclass antibodies, as shown in Figure 6B.

We also wondered whether any anti-carrier protein (BSA or CRM197) antibodies were generated. To prove that, the ELISA plates were coated with BSA or CRM197, and the sera were tested; no antibody was found, proving the success of the strategy.

## 4. Discussion

The development of effective anti-nicotine vaccines to fight smoking addiction has proven challenging. In recent years, mRNA vaccines have shown great potential against infectious diseases. However, for those that are not protein/peptide antigens, which could be the direct products of mRNA transcription, mRNA vaccine design is difficult and cannot ensure accurate antigen production. The biosynthesis of such antigens, like polysaccharides, nicotine, or other secondary metabolites, may involve multiple enzymes, which increases the complexity and uncertainty of mRNA vaccines. Traditional vaccine construction methods are still important for these antigens. As an effective vaccine design strategy, protein conjugation has resolved the problems caused by the weak immunogenicity of many haptens and is still a promising approach to establishing an anti-nicotine vaccine candidate. Chemically, traditional hapten–protein conjugation usually involves amino-carboxylic acid condensation. Since most proteins are hydrophilic, condensation reactions would be better conducted in an aqueous solution or a mixed solution with water as the majority and organic solvents as additives. Some hydrophobic haptens like nicotine have poor solubility in this kind of environment, which may make the conjugation process slow. After conjugation, the physical property of the hapten-loaded protein is different from the native protein, which may change the conformation of the protein carrier. In fact, when we think about the fundamental mechanism of immunology, the three-dimensional structure of the protein will be destroyed while being digested by APCs.

Our strategy was mainly established just based on the fact that the only unchangeable structure is the peptide sequence. By using reductants, e.g., phosphine, to cut off the disulfides and organic solvents, e.g., DMF, and thus break down the intramolecular hydrogen bonds, proteins were unfolded to become expanded and flexible peptide strings. When conjugation was carried out between this kind of “strings” with haptens, the reaction rates were supposed to be improved, and some much more hindered conjugation sites, usually inaccessible in well-folded proteins, were found in this case. The more exposed peptide backbone of the string-like structure also makes the whole molecule easier to be solvated by organic solvents like DMF so that conjugation could be carried out in an organic solvent, where some hydrophobic haptens like nicotine can be well dissolved, resulting in accelerated conjugation. Moreover, we utilized the highly chemically selective thiol–maleimide “click” reaction [23] instead of amino-carboxylic acid condensation for conjugation to avoid the possible protein–protein coupling. Additionally, the reaction between thiol and maleimide is much faster than amino-carboxylic acid condensation, further speeding up the conjugation process. In our protocol, it took only up to 5 h to obtain the thiol–maleimide conjugates, i.e., CRM197-Nicotine(4), BSA-Nicotine(24), and BSA-Nicotine(35), whereas to prepare CRM197-Nicotine(18) and BSA-Nicotine(77), it took at least 20 h. Although it is a common occurrence that the thiosuccinimide linkage may be eliminated through a retro-Michael pathway or thiol exchange in the presence of thiol-containing compounds, the Cao group successfully solved this problem by treating the thiosuccinimide with an alkaline solution in the aqueous phase [24,25,26,27]. The approved anti-Hib vaccine QUIMI-HIB^®^ contains thiosuccinimide as well [28,29]. Thus, the stability and safety of the maleimide–thiol adducts should not be a concern.

Besides the successful and practicable chemical process, we wondered, in terms of immunogenicity, if the conjugates prepared by thiol–maleimide addition in the organic phase were comparable with those obtained from traditional condensation in the aqueous phase. We fully formulated six vaccines to investigate their immunological efficacy (Table 1). During the vaccination protocol, no significant decreases in weight were observed, preliminarily demonstrating the safety of these vaccine candidates (Figure 4). The total IgG titer in mice induced by CRM197-Nicotine(4) was not obviously observed even after the third immunization, which is not surprising due to the too-low nicotine loading number. The IgG titers generated in the mice vaccinated with the other two thiol–maleimide conjugates, i.e., BSA-Nicotine(24) and BSA-Nicotine(35), were of low degrees after the first immunization (Figure 4) but kept rising rapidly during the following boost immunization and finally came to a comparable level with the IgG titers in the mice immunized with CRM197-Nicotine(18) and BSA-Nicotine(77), which were prepared using the traditional approach. Despite the fact that the immunogenicity of CRM197 is much higher than that of BSA, BSA-based conjugate BSA-Nicotine(24) elicited as many IgG antibodies after the final immunization. This result encouraged us because it demonstrated the potential that assembling a hapten–protein conjugate through our strategy may elicit a similar immune response even using the less expensive and less immunogenic BSA. This is economically friendly not only for academia but also for the industry.

It should also be noted that the IgG titers stimulated with BSA-Nicotine(77) after the third injection did not significantly increase compared to the second injection, although the immune response to this conjugate seemed to be initially stronger than the less nicotine-loaded BSA-Nicotine(24) and BSA-Nicotine(35). This suggests that, to construct hapten–protein vaccines, it may be not suitable to increase the loading number unlimitedly. Some anti-nicotine vaccine studies reported a similar phenomenon [4]. The contribution of the adjuvants was also significant. From the parallel comparison of the IgG titers induced by BSA-Nicotine(77)/Al and BSA-Nicotine(77), it was found that IgG antibodies in BSA-Nicotine(77)/Al treated mice were always much fewer than those in the mice vaccinated with BSA-Nicotine(77) formulated with MPL/QS-21 after each immunization. In fact, they were even fewer than the IgG titers the less nicotine-loaded BSA-Nicotine(24) and BSA-Nicotine(35). These results indicate that MPL/QS-21 would be a more promising adjuvant in anti-nicotine vaccine formulation than Al(OH)_3_. This is within our expectations. The adjuvant AS01, a liposome-based system containing MPL and QS-21, has been developed for nearly three decades and has already been approved for GSK’s malaria (Mosquirix) and shingles (Shingrix) vaccines [20,30]. To the best of our knowledge, no anti-nicotine vaccine studies have used MPL/QS-21 as the adjuvant.

IgG subclasses of these conjugates were investigated (Figure 6). The mice stimulated with BSA-Nicotine(24), BSA-Nicotine(35), and CRM197-Nicotine(18) all produced predominantly IgG1 antibodies. However, the IgG2a levels of mice immunized with BSA-Nicotine(24), BSA-Nicotine(24), and CRM197-Nicotine(18) should not be neglected (Figure 6A,B). The immune response was a Th1/Th2 mixed type [20,30]. By contrast, BSA-Nicotine(35) elicited very few IgG2a antibodies, indicating that the immune response to BSA-Nicotine(35) was Th2-bias. However, to generate prolonged protection, a Th1 response is still necessary. Meanwhile, as shown in Figure 5C, the total IgG titer induced by BSA-Nicotine(35) was at a significantly lower level than BSA-Nicotine(24), indicating that the latter may be more promising. At the current stage, we attribute the reason why BSA-Nicotine(35) seems relatively less effective than BSA-Nicotine(24) to the loading number of nicotine. Detailed studies on the influence of hapten loading may give us some insight into this factor. Figure 6E shows the titers of IgG subclasses induced by the aluminum-adjuvant-formulated BSA-Nicotine(77). Not surprisingly, they are all at low levels compared with the MPL/QS-21-adjuvant-formulated vaccines, once again demonstrating the robust immunostimulating function of this combination.

In conclusion, we established a novel approach to coupling haptens with unfolding protein through thiol–maleimide click chemistry in organic solvents for vaccine development. BSA-based nicotine conjugates were successfully assembled in this way, and BSA-Nicotine(24) induced a robust immune response with a Th1/Th2 mixed pattern and demonstrated comparable immunological efficacy with the more immunogenic carrier CRM197-based conjugate. Meanwhile, the powerful immunological enhancement function of MPL/QS-21 adjuvants was proven. Further studies on the structural improvement of the BSA-nicotine conjugates and the application of our conjugation strategy on the design and synthesis of other kinds of protein-carried antigens are currently underway in our laboratory.

## Data Availability

All data generated or analyzed during this study are included in this published article and its Appendix A.

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
