# Peer review of "Unfolding Protein-Based Hapten Coupling via Thiol–Maleimide Click Chemistry: Enhanced Immunogenicity in Anti-Nicotine Vaccines Based on a Novel Conjugation Method and MPL/QS-21 Adjuvants"

_polymers, 2024, doi:10.3390/polym16070931_

Round 1
Reviewer 1 Report
Comments and Suggestions for Authors
Comments:
1. What is the advantage of this developed vaccines compared to the current mRNA vaccine? The author should provide more comments on that.
2. The authors have quantified the IgG tiers for evaluating the vaccine functionality, I was wondering if they have quantified the CD4 or CD8 T cell response for their study.
Reviewer 2 Report
Comments and Suggestions for Authors
· The manuscript presents a detailed investigation into the development of anti-nicotine vaccines using a novel protein conjugation strategy.
· The approach seemed successful since it produced immune response in mice against nicotine.
· The safety profile of the vaccines was established.
· The scope and limitations of the study were addressed.
· A plausible mechanism of action, involving the unfolding of proteins to expose conjugation sites, was provided.
· The role/effect of adjuvants was investigated.
· The use of BSA as a cheaper alternative to CRM197 was explored, at the same time the need for multiple booster doses to achieve comparable efficacy was reported.
1. What is the main question addressed by the research? The main question addressed by the research is the development of effective anti-nicotine vaccines to combat smoking addiction, particularly focusing on conjugating maleimide-modified nicotine hapten with a disulfide bond-reduced carrier protein in organic solvent to enhance immunogenicity. 2. What parts do you consider original or relevant for the field? What specific gap in the field does the paper address? 3. What does it add to the subject area compared with other published material? A novel conjugation strategy, aiming to overcome the challenge of weak immunogenicity associated with small molecules like nicotine; and the use of BSA as a cheaper alternative. The proposed method also offers advantages over traditional carbodiimide crosslinking chemistry, including enhanced flexibility of peptide conformation and increased exposure of conjugation sites. The study also investigates the effects of different adjuvants on vaccine efficacy, providing valuable insights for further vaccine development studies. 4. What specific improvements should the authors consider regarding the methodology? What further controls should be considered? All experimental designs, controls, and protocols are adequate. 5. Please describe how the conclusions are or are not consistent with the evidence and arguments presented. Please also indicate if all main questions posed were addressed and by which specific experiments. The conclusions drawn from the evidence and arguments presented in the paper are consistent. The main questions posed regarding the development and immunogenicity of anti-nicotine vaccines were addressed through experiments involving the vaccination of mice with different vaccine formulations. 6. Are the references appropriate? The references provided in the paper appear to be appropriate, covering relevant literature on vaccine development, conjugation chemistry, and adjuvant usage in immunology. 7. Please include any additional comments on the tables and figures and quality of the data.The tables and figures included in the paper appear to be of good quality and effectively illustrate the experimental results and findings.
Small suggestion:
Proof-read the manuscript again to correct for minor mistakes/repetitions, and please introduce the missing abbreviations.
